# Commissural Alignment and Coronary Access after Transcatheter Aortic Valve Replacement

**DOI:** 10.3390/jcm12062136

**Published:** 2023-03-09

**Authors:** Angelo Quagliana, Nicholas J. Montarello, Yannick Willemen, Pernille S. Bække, Troels H. Jørgensen, Ole De Backer, Lars Sondergaard

**Affiliations:** 1The Heart Center, Rigshospitalet, Copenhagen University Hospital, 2100 Copenhagen, Denmark; 2Cardiocentro Ticino Institute—EOC, Universita’della Svizzera Italiana, 6900 Lugano, Switzerland

**Keywords:** transcatheter aortic valve replacement, transcatheter aortic valve implantation, commissural alignment, coronary access

## Abstract

Transcatheter aortic valve implantation (TAVR) is the first therapeutic option for elderly patients with severe symptomatic aortic stenosis, and indications are steadily expanding to younger patients and subjects with lower surgical risk and longer life expectancy. Commissural alignment between native and transcatheter valves facilitates coronary access after TAVR and is thus considered a procedural goal, allowing long-term management of coronary artery disease. Moreover, commissural alignment may potentially have a positive impact on transvalvular hemodynamic and valve durability. This review focus on technical hints to achieve commissural alignment and current evidence for different transcatheter aortic valves.

## 1. Introduction

Transcatheter aortic valve replacement (TAVR) has become the standard treatment for elderly patients with severe, symptomatic aortic stenosis (AS) across all surgical risk strata. More recently, the indications for TAVR have expanded to include a younger patient cohort with fewer comorbidities.

### 1.1. Prevalence of Coronary Artery Disease in Patients Undergoing TAVR

Coronary artery disease (CAD) frequently occurs concomitantly with AS due to shared predisposing factors and pathophysiology [1]. Approximately 50% of patients with severe symptomatic aortic stenosis undergoing TAVR have co-existent CAD [2], although definitions are often inconsistent across studies, and its probability is highly related to the surgical risk profile, ranging between 30% and 75% [3,4,5]. Patients at low surgical risk are less likely to present with symptomatic coronary atherosclerosis, but the incidence of coronary events is anticipated to increase with age due to the progressive nature of the disease [1], and subsequent coronary angiography (CAG) and percutaneous coronary intervention (PCI) may be needed, even in younger AS subjects referred for transcatheter treatment.

### 1.2. CAG and PCI after TAVR

Recent studies have reported an incidence of acute coronary syndrome (ACS) following TAVR in the order of 5% at a median follow-up of 1 year [6,7] and 10% at a median follow-up of 2 years [8]. Further, at 8 years follow-up, the NOTION-1 trial demonstrated a cumulative incidence of acute myocardial infarction of 6.2% in the TAVR cohort [9]. The predominant ACS presentation in TAVR patients is non-ST-segment-elevation myocardial infarction (NSTEMI), which is in accordance with the higher frequency of NSTEMI seen in increasingly elderly patients [8]. It has been suggested that the lower proportion of ST-segment-elevation myocardial infarction (STEMI) may relate to post-TAVR anti-thrombotic pharmacotherapy [6]. Post-TAVR STEMI patients have a higher risk of major adverse cardiovascular and cerebrovascular events (MACCE) and increased mortality both in-hospital and following discharge, compared to non-TAVR STEMI [6,10]. Thus, Faroux et al. demonstrated longer door-to-balloon time and higher failure rates in primary PCI after TAVR [10], which was associated with increased mortality and MACCE. This often relates to the complexity of coronary cannulation after transcatheter aortic valve replacement in an acute setting and emphasizes the importance of improving coronary artery accessibility.

### 1.3. Difficulties in Coronary Access after TAVR in Relation to Valve Design and Implantation Technique

The ease of coronary engagement after TAVR depends on an interplay of several factors related to the native anatomy of the patient, the transcatheter heart valve (THV) design, and implantation characteristics (summarized in Table 1).

With respect to the patient's anatomy, coronary cannulation is normally more challenging when the coronary ostia are located low within the coronary sinus and when the aortic root is small, with reduced maneuvering space for angiographic catheters.

Figure 1 illustrates four widely used THVs and the most important features of their design that may impact coronary access. A tall THV stent frame covering the SOV is a potential obstacle to coronary cannulation compared to a lower stent frame [11], especially when the THV has a small stent cell design. A taller outer skirt, designed to prevent paravalvular leak (PVL), may also make coronary engagement more difficult [12]. Skirt interference can be more problematic when the THV is implanted in a high position to reduce the incidence of conduction disorders and to mitigate the risk of PVL [11]. Further, supra-annular leaflet designs are more likely to cover coronary ostia during valve opening, reducing the likelihood of successful coronary engagement [11]. Pre-procedural planning based on computer tomography (CT) measurements is therefore of paramount importance, as it allows the selection of the most suitable THV and its optimal implantation depth to facilitate future coronary access.

The RE-ACCESS study determined that a high transcatheter aortic valve/sinus of Valsalva ratio, THV implantation depth, and use of Evolut devices were predictors of difficult coronary cannulation after TAVR [13]. In addition, in a sub-study analysis of the RESOLVE registry comparing Evolut R/Pro (Medtronic, Minneapolis, MN, USA) with SAPIEN valves (Edwards Lifesciences, Irvine, CA, USA), Ochiai et al. noted unfavorable coronary access features for both the LCA and RCA in 34.8% and 25.8%, respectively, of patients treated with the Evolut THV, compared with 15.7% and 8.1%, respectively, in the SAPIEN THV group [11]. 

More recently, the rotational orientation of the device after deployment has been described as a factor influencing future coronary cannulation possibilities. When THV and native commissures are not overlapped (i.e., in cases of commissural misalignment, CMA), neo-commissures may face coronary ostia and create a potential anatomical barrier to future engagement. 

Accordingly, improvement in THVs and delivery systems to help achieve commissural alignment are nowadays at the forefront of attempts to optimize long-term management of coronary artery disease after TAVR.

## 2. Delivery System Technologies and Impact on Commissure Orientation

Tang et al. evaluated the impact of the initial THV deployment orientation on neo-commissural alignment and overlap with coronary arteries [14]. In the Alignment of Transcatheter Aortic-Valve Neo-Commissures (ALIGN-TAVR) study, 828 TAVR implants (SAPIEN 3, *n* = 483; Evolut, *n* = 245; ACURATE neo, *n* = 100) were evaluated using pre-procedural multidetector row CT and post-implant co-planar fluoroscopy co-registration. The Evolut THV platform has one of the commissures orientated in the same direction as the C-paddle, which is at an angle of 90° to the “Hat” marker, Figure 2.

The cohort with “Hat” on the outer-curve (OC) or center-front (CF) of the aortic annulus prior to THV deployment had improved commissural alignment and a lower incidence of coronary overlap compared those with “Hat” on the inner curve (IC) and center back (CB) (36% vs. 60%, *p* < 0.05). Orientation of the flush port of the delivery system at three o’clock (pointing towards the left of the patient) on insertion into the right femoral artery achieved a desired “Hat” in 91.6% of cases. The corresponding orientation of the flush port for the ACURATE neo2 THV (Boston Scientific, Marlborough, MA, USA) is six o’clock, and for the Navitor THV (Abbott, Abbott Park, IL, USA) is 12 o’clock. Unlike the self-expandable platforms, however, the initial crimped orientation of the SAPIEN 3 THV, consisting of three fluorogenic commissural tabs with four struts in between, had no impact on the final valve orientation.

Despite the improvement seen in commissural alignment following specific delivery system orientation at insertion to the common right femoral artery, the risk of severe coronary overlap remained high (24.3% for Evolut), reflecting the diversity in patient anatomies [14]. Consequently, there has been increasing interest in the utilization of patient-specific implantation techniques to further mitigate the risk of commissural misalignment.

### 2.1. Patient-Specific Implantation Techniques to Obtain Neo-Commissural Alignment

#### 2.1.1. Self-Expanding THV Platforms

Bieliauskas et al. recently proposed a fluoroscopy-based, patient- and device-specific TAVR implantation technique for several self-expanding THV platforms [15]. In the Commissural Alignment (COMALIGN) study, pre-procedural CT analysis was utilized to determine the patient-specific fluoroscopic projection (or C-arm projection) in which the right (RCC) and left coronary cusps (LCC) were overlapping (cusp overlap view). This patient-specific fluoroscopic projection was then used to achieve neo-commissural alignment, with the recognition that in cusp overlap view, the native RCC/LCC commissure is directed to the right of the fluoroscopic image. Subsequent orientation of one of the THV commissures to the right of the screen in this same view was achieved, utilizing the unique fluoroscopic markers seen on each of the THVs or their delivery catheter (Figure 2).

After insertion of the system with the flush port at three o’clock for Evolut THV, six o’clock for ACURATE neo2 THV, and 12 o’clock for Navitor THV, a consistent implant sequence was followed. On reaching the level of the native annulus, the Evolut THV was slowly torqued until the “Hat” marker was in the CF position and the C-paddle was to the right of the screen in the cusp overlap view before THV deployment. Thereby, the THV commissures are aligned with the commissures of the native valve. For the ACURATE neo2 THV, the commissures are located at the same position as the posts, and the ‘free stent struts’ are visible on fluoroscopy. The commissure posts located mid-stent frame are less visible on the current generation Navitor THV. Post-TAVR CT assessment revealed that of 60 evenly distributed patients across these three platforms, optimal alignment (CMA < 15°) or mild commissural misalignment (CMA < 30°) was achieved in 53 patients (88%). No moderate or severe commissural misalignment was seen with the ACURATE Neo2 platform due to clearly visible landmarks for the commissures and easy torque of the delivery system.

Kitamura et al. readapted the same technique on patients with failed surgical bioprosthetic valves undergoing valve-in-valve treatment with Evolut platforms [16]. A preliminary CT scan was used to determine the fluoroscopic view wherein the surgical commissural post between the left and the right coronary cusp is isolated at the right of the screen (RCC-LCC overlap view). According to the previously described sequence, the delivery catheter was inserted with the flush port at three o’clock and advanced up to the ascending aorta, keeping the hat marker on the outer aortic curve. THV rotational orientation was then corrected above the annular plane in cusp overlap view until the Hat marker reached a CF position with the C-Tab pointing right (towards the isolated surgical post). Subsequent valve deployment successfully resulted in good commissural alignment (CMA < 30°) in 95% of 13 consecutive patients.

De Marco et al. used an alternative patient-specific algorithm to achieve commissural alignment with the ACURATE neo THV [17]. The sextant technique aims to position the THV so that one of the commissural posts lies upon the internal bisector of the angle between the coronary arteries assessed by pre-TAVR CT, with a rotational error of less than 15% in either direction. A dedicated fluoroscopic view 30° from the bisector towards the right coronary artery is established. Correct alignment of the THV device is recognized when the commissural post appearing as a line lies on the left of the screen and the two angled commissural posts are superimposed. If the orientation is not correct, the delivery system is retracted into the descending aorta, rotated 30° in a stepwise fashion and re-advanced. Using this technique, the THV was aligned or only mildly misaligned in 42 of 45 patients.

Other self-expandable THV platforms are available now, and similar workflows to achieve commissural alignment during implantation have been described. Hydra THV (Sahajanand Medical Technologies, Wakhariawadi, India) has three wide, tentacle-shaped arches in the outflow tract, with three commissural posts rising up in between. Six radiopaque markers in two rows are located on the stent frame, to guide valve positioning. These markers highlight the center points of the three THV cusps. To achieve commissural alignment, two markers should be overlapped at the right of the screen in RCC-LCC cusp overlap view, isolating one marker on the left. A “near 1:2 pattern” has been shown to result in good CA and easy coronary engagement after valve positioning [18], while a perfect 1:2 configuration is not easy to obtain for a slightly asymmetric distance between radiopaque markers.

#### 2.1.2. Balloon-Expandable SAPIEN 3 Platform

To date, there is no documented method on how to achieve commissural alignment, with the consequence that 33–43% of SAPIEN 3 THVs are deployed with moderate or severe misalignment [19].

## 3. Commissural vs. Coronary Ostial Alignment

Coronary access may be difficult or unachievable following TAVR if the THV commissures are close to the coronary ostia, but the attainment of patient-specific commissural alignment during self-expandable THV deployment can reduce the risk of unsuccessful coronary access to below 5% [13,20]. However, commissural alignment alone will not always yield an optimal THV orientation for coronary engagement, as other factors may play a role. Notably, cusp symmetry of the native aortic valve influences coronary access after TAVR, as any asymmetry will impair commissural alignment given that the angle between two commissures of the THV is always precisely 120°. Recently, the ALIGN-TAVR Consortium proposed a standardized definition with regard to what constituted native cusp symmetry, based on the inter-commissural angle of the largest cusp (Figure 3A) [21]. When this angle is between 120 and 125°, the valve is deemed symmetric; angles greater than 125° define mildly to severely asymmetric valves with each 5° increment. Using this criteria, cardiac CT evaluation of 200 tricuspid and 200 bicuspid aortic valves showed that severe cusp asymmetry was more frequent in BAVs (52.5%) than in TAVs (2.5%) (*p* < 0.001), with the non-coronary cusp being the most common dominant cusp [22]. Coronary ostium eccentricity is another factor that affects the risk of coronary/neo-commissure overlap following TAVR, despite optimal commissural alignment. The ALIGN-TAVR Consortium also proposed that the following standardized definition of coronary ostium eccentricity be used, based on the angle between the coronary ostium and the center of the respective cusp: 0°–10° defines centered, 10°–20° mild coronary ostia eccentricity, 20°–30° moderate coronary ostia eccentricity, and >30° severe coronary ostia eccentricity (Figure 3B) [21].

Cardiac CT analysis of the aortic valves revealed that the RCA more often has an eccentric take-off compared to the LCA [22]. Thus, the RCA ostium may be located closer to the commissure between the right and the non-coronary cusp, with 28% of patients having more than mild RCA eccentricity, compared to 6% for LCAs (*p* < 0.001). These data raise concerns about potentially difficult coronary cannulation in a large group of patients, despite optimal commissural alignment of the THV, because of cusp asymmetry or coronary eccentricity. In a virtual CT-based THV simulation on 100 tricuspid aortic valves, the optimal commissural alignment was compared to the optimal coronary alignment through measurement of the final distance between the coronary ostia and THV commissures (coronary overlap) [23]. To obtain coronary alignment, the distance from the neo-commissures to the coronary ostia was maximized by positioning one of the neo-commissures at the bisector of the angle between the ostia of the RCA and LCA. Coronary overlap was defined as the smallest angle between a neo-commissure and a coronary ostium: 35°–60° indicated no overlap, 20°–35° moderate overlap, and ≤20° severe overlap. In simulations with optimal commissural alignment, the rate of severe and moderate overlap was 5% and 27%, respectively, predominantly involving the RCA (29 out of 32 cases). There was no severe overlap, and only 5% moderate overlap in simulations with optimal coronary alignment. These findings suggest that it is more beneficial to achieve coronary alignment over commissural alignment for coronary accessibility following TAVR. Furthermore, two features were identified as increasing the risk of coronary overlap with THV commissures: (1) an extreme angle between LCA and RCA (<103° and >147°), and (2) coronary ostium eccentricity within the sinus of Valsalva of >27° for the RCA and >19° for the LCA.

Commissural alignment is normally achieved by using a fluoroscopic right–left cusp overlap view for orientation of the THV during implantation. Alternatively, to achieve coronary alignment, cardiac CT can determine the angiographic view wherein coronary ostia, rather than cusps, overlap at the annular level (Figure 3C) [22]. Orienting the THV using this view during valve deployment so that one of the commissural posts lines up at the right of the angiographic image would ideally lead to one of the neo-commissures landing equidistantly between the RCA and LCA ostia, therefore optimizing coronary re-access.

Notably, in the study by Wang et al., the right–left cusp overlap view and coronary ostia overlap view differed by less than 10° of fluoroscopic angulation in three out of four cases, upon CT analysis of 200 tricuspid and 200 bicuspid aortic valves [22]. Only 2.5% of cases demonstrated a difference in fluoroscopic angulation of 20° or more, in either the LAO/RAO or cranial/caudal direction. In 97% of tricuspid and 93% of bicuspid aortic valves, the RCC-LCC centered line and RCA-LCA bisector deviated by less than 20°, implying that in most cases, a patient-specific implantation technique aimed at obtaining commissural alignment will also effectively yield coronary alignment. 

In specific cases with marked cusp asymmetry and/or coronary ostium eccentricity, a coronary ostia overlap view instead of a cusp overlap view could therefore be used during TAVR in order to position one of the neo-commissures centrally between the RCA and LCA [22]. This strategy decreases the risk of coronary overlap with one of the commissural posts of the THV in virtual simulation [23]. The clinical impact and relevance of coronary access following TAVR, however, have not been investigated and remain uncertain. Moreover, implantation that is focused on coronary alignment may also have some drawbacks: (1) it could improve access to the RCA but, paradoxically, make access to the LCA more difficult, which could be problematic or possibly detrimental incases of left coronary dominance, (2) implantation views to obtain coronary alignment could require more extreme and unfeasible C-arm angulations compared to cusp overlap view, and (3) commissural alignment may potentially have a positive impact on THV function and durability. Sacrificing this in favor of coronary alignment requires further consideration. Perhaps only in a minority of patients would extra attention to coronary ostia and a patient-specific coronary alignment implantation technique be justified to optimize future coronary access.

## 4. Possible Implications of Commissural Alignment beyond Coronary Access: Lights and Shadows

Facilitating future coronary access has highlighted the importance of commissural alignment during THV positioning Apart from the long-term management of coronary artery disease, the percutaneous treatment of younger patients raises concerns about valve durability, and strategies to optimize THV implantation, improve transcatheter valve function, prevent valve deterioration, and allow for THV-in-THV procedures are now paramount. 

The geometry of THVs after deployment plays a leading role in valve expansion and functionality. Whether the rotational orientation of a prosthetic device can impact the morphology and affect the physiology of THV leaflets is still a concern, with possible implications for valve durability. 

A rationale to investigate the effects of misalignment on THV hemodynamics was first provided by Bailey et al., who showed on a computational model that the rotational orientation of the device can affect neo-leaflet expansion [24]. In their experiments, the deployment of an Edwards Sapien XT valve was repeatedly simulated on an aortic root model with different rotational orientations, ranging from commissural alignment to severe commissural misalignment. Moderate to severe CMA resulted in a shorter distance between prosthetic commissures due to stent frame adaptation to calcification of the native leaflets. This produced higher leaflet distortion, with an increase in trans-valvular shear stress. 

Following this, clinical observational experiences provided controversial results on the possible correlation between commissural alignment, trans-valvular flow conditions and clinical outcomes.

### 4.1. Commissural Alignment and Aortic Regurgitation

In 212 TAVR patients enrolled in the SAVORY and the RESOLVE registries, moderate or greater misalignment was shown to be an independent predictor of mild central aortic regurgitation at discharge for both self- and balloon-expandable THVs. Higher rates of central leaks for misaligned valves were confirmed at 3 month follow-up [25]. The CMA-related leaflet distortion shown in Bailey’s models could support this observation, though it has not been confirmed by other studies [26].

### 4.2. Commissural Alignment and Valve Thrombosis

Subclinical leaflet thrombosis is a common finding in both surgical and transcatheter bioprosthetic aortic valves. Although it is uncertain whether this imaging phenomenon may lead to any clinical consequences, there are concerns that it may adversely affect valve durability and lead to thrombo-embolic events [27].

The coagulative cascade can be triggered by low-flow conditions inside the neo-sinuses, and the morphological features and wash-out of these spaces can theoretically be affected by the rotational orientation of the device. 

A post hoc sub-analysis of the Low-Risk TAVR (LTR) Trial compared patients showing CT signs of subclinical leaflet thrombosis 30 days after TAVR to those who showed preserved leaflet morphology in order to identify possible predictors of this imaging phenomenon [28]. Among cases with leaflet thickening, the prevalence of commissural misalignment was higher compared to controls (40% vs. 28%). In another prospective CT evaluation at 30 days, in 512 TAVR patients demonstrating an incidence of subclinical leaflet thrombosis ranging between 16 and 21% (for Evolut and Sapien 3 THVs, respectively), leaflet thickening was related to device deformation, asymmetry and to a small neo-sinus volume, but not to commissural misalignment [29]. 

These contradictory findings about a possible association between commissural alignment and leaflet thrombosis still require further investigation.

### 4.3. Commissural Alignment and Structural Valve Deterioration

Recently, the impact of commissural alignment on valve hemodynamic and related clinical outcomes has been retrospectively analyzed in a population of 324 consecutive patients treated with balloon-expandable THVs [26]. No overt effect of misalignment on transvalvular gradients was noted at discharge. As a possible signal of early flow alterations, however, misalignment was independently associated with higher rates of relative gradient increase >50% at 30 days, which was previously described as a predictor of thrombosis for surgical bioprosthetic aortic valves [25]. No associations between misalignment and VARC-2-defined prosthetic dysfunction, central or paravalvular regurgitation or rates of subclinical leaflet thrombosis were found, however; this contradicts previous observations.

### 4.4. Commissural Alignment and THV-in-THV

As TAVR is being offered to younger patients whose life expectancy may exceed THV durability, redo TAVR procedures are expected to become more common in the future. This possibility deserves consideration and should influence initial valve choice and deployment technique. 

In redo TAVI, the new valve displaces the degenerated leaflets of the first THV between two stent frame layers. This creates a “neo-skirt” that can reach sinotubular junction (STJ) level and result in sinus sequestration, with abrupt and life-threatening impairment of coronary flow. This risk is higher if the distance between the stent frame of the first valve and the STJ is short, especially in degenerated THVs with supra-annular leaflet design in which the neo-skirt may cover the SOV entirely [30]. In redo TAVR with a high risk of sinus sequestration, intentional leaflet modification (e.g., BASILICA) of the index THV leaflets to prevent iatrogenic coronary artery obstruction has been proposed, since a lacerated neo-skirt is less likely to exclude sinuses from being perfused. However, leaflet laceration in THV-in-THV procedures may only be effective if the index THV has commissural or coronary alignment. As a result, redo TAVR in a misaligned index THV has an intrinsic higher risk of coronary perfusion impairment which cannot be efficiently corrected by currently available strategies.

## 5. Conclusions

Indications for TAVR have steadily expanded over the last decade, and nowadays include most elderly patients with severe AS across the entire spectrum of surgical risk. Percutaneous treatment of younger subjects with longer life expectancy poses new challenges to transcatheter therapy in which lifetime management is a predominant concern and optimizing clinical outcomes is mandatory. Optimization has been focused on several aspects of THV design and the implantation procedure. 

Commissural alignment has gained attention since observation experiences have shown easier coronary re-access and safe redo TAVR in cases of valve failure. A possible impact on valve durability and subclinical leaflet thrombosis needs further investigation.

## Figures and Tables

**Figure 1 jcm-12-02136-f001:**
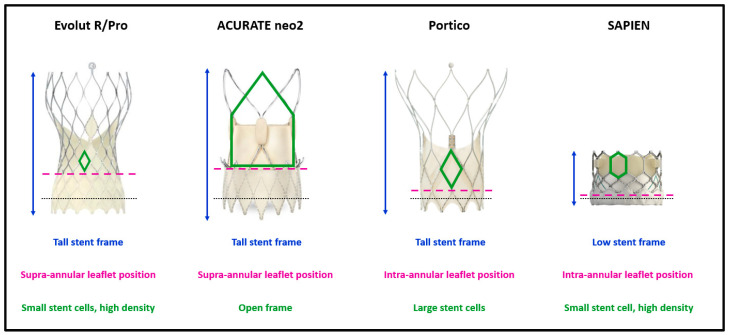
THV characteristics impacting coronary access after TAVR. Black dotted lines remark the level of native annulus at nominal implantation depth. THV = transcatheter heart valve. TAVR = transcatheter aortic valve replacement.

**Figure 2 jcm-12-02136-f002:**
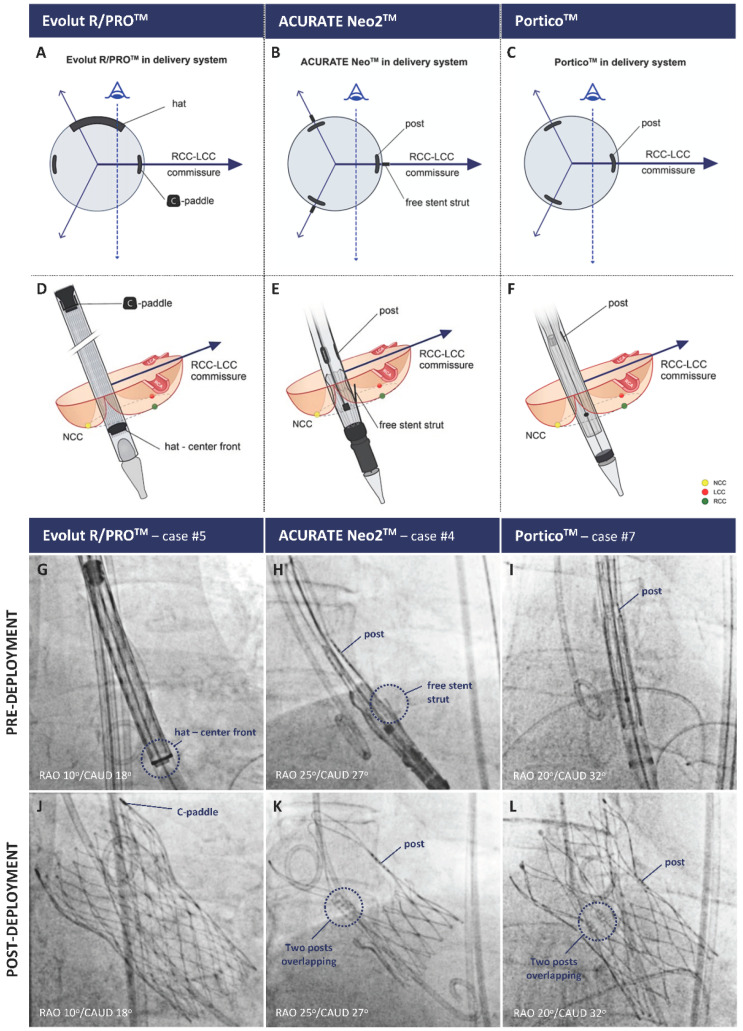
Positioning of the radiopaque THV commissural markers to obtain neo-commissural alignment using the RCC/LCC cusp overlap view for the Evolut R/PRO+ (**left column**), ACURATE neo2 (**middle column**), and Navitor platforms (**right column**). Panels (**A**–**C**), radiopaque marker orientation in a transversal plane; Panels (**D**–**F**), schematics of radiopaque marker positioning on fluoroscopy; Panels (**G**–**L**), fluoroscopic images before and after THV deployment, respectively. THV = transcatheter heart valve. LCC = left coronary cusp. NCC = non-coronary cusp. RCC = right coronary cusp. Reprinted from Bieliauskas et al. [15].

**Figure 3 jcm-12-02136-f003:**
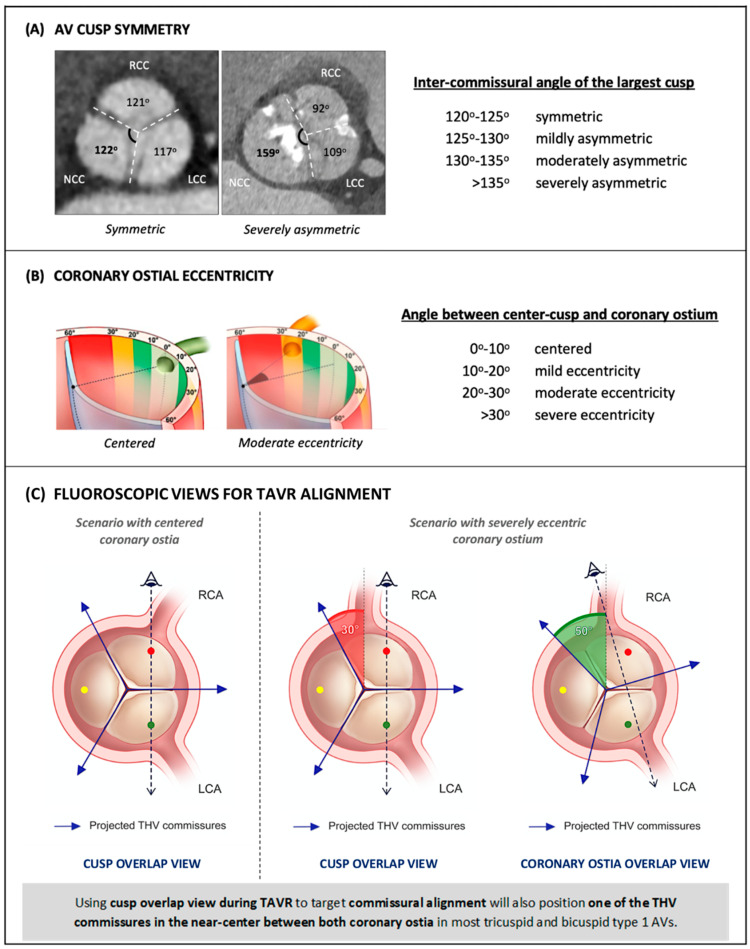
Proposed standardized definitions of aortic valve cusp symmetry (**A**) and coronary ostial eccentricity (**B**). (**C**): In the first scenario, on the left, commissural alignment results in an optimal angle between THV commissures and coronary ostia (60 degrees). In the second scenario, perfect commissural alignment results in a short RCA-to-commissure distance (30 degrees) due to RCA eccentricity. Using coronary ostia overlap view optimizes the commissure position in between, thereby facilitating RCA cannulation after TAVR. THV = transcatheter heart valve. RCA = right coronary artery. TAVR = transcatheter aortic valve replacement. AV = aortic valve. BAV = bicuspid aortic valve. LCA = left coronary cusp. NCC = non-coronary cusp. R/L = right-left. TAV = tricuspid aortic valve. Modified from Tang et al. [20] and Wang et al. [21].

**Table 1 jcm-12-02136-t001:** Factors impacting coronary re-access with respect to patient´s anatomy, THV design and implantation procedure. THV = transcatheter heart valve. SOV = sinus of Valsalva.

Factors Restraining Coronary Access	
**Native anatomy**	1. Coronary ostia
	Lower coronary take-off
	Ostial eccentricity
	2. Aortic root dimensions
	Lower sino-tubular junction and smaller diameter
	Smaller sinus of Valsalva diameter and area
**THV design**	1. Taller stent frame
	2. Higher sealing skirt
	3. Higher stent cell density
	4. Supra-annular leaflet positioning
**Procedural characteristics**	1. High THV implantation
	2. Commissural misalignment
	3. Greater THV-SOV relation

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
