# Peer review of "Commissural Alignment and Coronary Access after Transcatheter Aortic Valve Replacement"

_jcm, 2023, doi:10.3390/jcm12062136_

Round 1

Reviewer 1 Report

I read with interest the manuscript jcm-2216121 entitled "Commissural alignment and coronary access after transcatheter aortic valve implantation" by Dr Quagliana et al. The manuscript is well written and authors have to be commended for offering interesting technical insights on TAVI procedures. 

I have few comments:

- Few lines might be added on other existing and upcoming valves on the market.

- Also a comment on the existing valve-in-valve procedures might be of interest clarifying pro- and cons- of each strategy.

Author Response

Thanks for your interest in our review. We appreciated your suggestion and expanded our manuscript to include a new paragraph describing how patient-specific commissural alignment technique has been modified for valve-in-valve procedures with Evolut platforms and some considerations about CA technique with Hydra valves, to improve the didactic value of the paper. 

Reviewer 2 Report

In this exhaustive and and informative manuscript, Quagliana et al. review the potential role of commissural alignment during implantation of self-expanding valves during TAVI procedures in facilitating post-procedural coronary access. The manuscript is well-written and of interest to cardiologists who care for patients with aortic stenosis who are referred for TAVI. My one comment is that the data relating to the role of commisural alignment and improved coronary access is derived from observational studies and in-vitro modeling. The failure of cerebral embolic protection to reduce peri-procedural stroke rates in patients undergoing TAVI teaches us that pathophysiologically plausible assumptions do not necessarily translate into measurable clinical benefit. I therefore recommend that the authors state that the data relating to the role of commisural alignment in facilitating coronary access awaits validation in randomized clinical trials.

Author Response

Thanks a lot for your interest in our manuscript. Commissural alignment is indeed a trending topic and part of the strategy to optimize TAVI outcomes in a long-term perspective. We agree that what we know so far exclusively comes from observational experiences, which are summarized in our work. According to your comments we declare it more clearly in our conclusions, remarking the need for further evidences.